# Quantum spin-engineering in on-surface molecular ferrimagnets

Wantong Huang ®[1], Máté Stark[1], Paul Greule[1], Kwan Ho Au-Yeung ®[1], Daria Sostina[1,2], José Reina Gálvez ®[3,4], Christoph Sürgers ®[1], Wolfgang Wernsdorfer ®[1,2], Christoph Wolf ®[3,4] & Philip Willke ®[1,5] ✉

The design and control of atomic-scale spin structures constitute major challenges for spin-based quantum technology platforms, including quantum dots, color centers, and molecular spins. Here, we showcase a strategy for designing the quantum properties of molecular spin qubits by combining tip-assisted on-surface assembly with electron spin resonance scanning tunneling microscopy (ESR-STM): We fabricate magnetic dimer complexes that consist of an iron phthalocyanine (FePc) molecule and an organometallic half-sandwich complex formed by the FePc ligand and an attached iron atom, $Fe(C_6H_6)$. The total complex forms a mixed-spin (1/2,1) quantum ferrimagnet with a well-separated correlated ground state doublet, which we utilize for coherent control. As a result of the correlation, the quantum ferrimagnet shows an improved spin lifetime ( >1.5 µs) as it is partially protected against inelastic electron scattering. Lastly, the ferrimagnet units also enable inter-molecular coupling, that can be used to realize both ferromagnetic or anti-ferromagnetic structures. Thus, quantum ferrimagnets provide a versatile platform to improve coherent control in general and to study complex magnetic interactions.

Protecting individual qubits from interaction with the environment is one of the crucial challenges for quantum information processing. For various quantum architectures, a plethora of design strategies were developed that alter the properties of the system in a way that makes it resilient to various interactions. One prominent example constitutes the evolution of superconducting qubits, which transitioned from noisy charge qubits to transmon qubits[1], the latter featuring a more robust energy level landscape. Also, for spin-based quantum architectures, a variety of designs were employed, including clock transitions[2], singlet-triplet qubits in semiconductor quantum dots[3] or chirality-based quantum states[4]. In particular, for molecular spin qubits, even larger interacting systems have been proposed and realized[5–7]. Conventionally, these spin systems were enabled by chemical synthesis. An alternative bottom-up route for creating interacting spin structures is on-surface synthesis, monitored and assisted by

scanning tunneling microscopy (STM)[8]. This includes various interacting magnetic spin systems that were shown to form complex spin structures[9–16]. However, probing the intrinsic spin properties remains challenging and has up to now mostly been indirectly interrogated via the interaction with the substrate conduction electrons, i.e., the Kondo effect. A viable solution to this problem is to decouple the molecular spins from the metallic substrate via thin insulators[17–19]. This, however, makes it challenging for on-surface chemistry methods[8].

A direct way to probe spin properties and to obtain insight into their spin dynamics is to use electron spin resonance in a scanning tunneling microscope (ESR-STM)[20]. This allows for probing various spin systems, for instance transition metal atoms[21], alkali atoms[22], rare earth elements[23,24], as well as molecular spins[17,25,26]. Moreover, it permits to control the spin coherently in the time domain[27,28]. However, short spin lifetimes constitute a major challenge in most spin systems

[1]Physikalisches Institut, Karlsruhe Institute of Technology (KIT), Karlsruhe, Germany. [2]Institute for Quantum Materials and Technologies, Karlsruhe, Germany. [3]Center for Quantum Nanoscience, Institute for Basic Science (IBS), Seoul, Republic of Korea. [4]Ewha Womans University, Seoul, Republic of Korea. [5]Center for Integrated Quantum Science and Technology (IQST), Karlsruhe Institute of Technology, Karlsruhe, Germany. ✉e-mail: philip.willke@kit.edu

($T_1 < 300$ ns[17,25,28]), which also limits their phase coherence time $T_2 \leq 2T_1$. For $T_1$ times, the main limitation remains the scattering with nearby tunneling electrons emanating from the tip and substrate electron baths[27,29,30]. Thus, one viable strategy is to move to thicker layers of the underlying insulator magnesium oxide (MgO)[29], or to employ other routes such as utilizing atomic force microscopy[31]. An alternative strategy is to make the spin systems intrinsically more robust against sources of noise and relaxation by engineering their magnetic interactions.

In this work, we demonstrate how a tip-assisted assembly of a molecular complex leads to a spin system with improved dynamic spin properties compared to the constituents. The complex consists of an iron phthalocyanine (FePc) molecule and an additional Fe atom. The latter forms together with part of the FePc ligand an organometallic half-sandwich complex. We find that the resulting spin system constitutes a mixed-spin(1/2,1) Heisenberg quantum ferrimagnet[32]. The ferrimagnet has an improved spin lifetime compared to conventional on-surface spin ½ systems[27,28], reaching here up to $T_1 = 1.6$ µs. This increase is rationalized by employing spin transport calculations, which show that the inelastic scattering channels limiting $T_1$ are suppressed by the correlations in the quantum ferrimagnet. In addition, we demonstrate that multiple complexes can be efficiently coupled either ferromagnetically or antiferromagnetically depending on their relative alignment.

## Results

Figure 1a shows an STM topography of the sample system, that consists of individual Fe atoms and FePc molecules on two monolayers (ML) of MgO atop a Ag(001) substrate. FePc molecules were previously shown to be singly charged, resulting in a mostly isotropic spin $S = 1/2$ system[25,26], while individual Fe atoms are spin $S = 2$ systems with a large out-of-plane magnetic anisotropy $DS_z^2 (D = -4.6$ meV)[33]. From these building blocks, we create molecular complexes in which the two Fe atoms are strongly coupled. We employ a simple tip-assisted assembly scheme, in which an FePc molecule is picked up by the STM tip and subsequently dropped atop the Fe atom (Supplementary Fig. 1). In Fig. 1a we show several of these complexes as well as dimers of complexes (discussed in Fig. 4). Lattice site analysis (Supplementary Fig. 2) as well as density functional theory (DFT) calculations (Fig. 1b and Supplementary Fig. 2b) show that both Fe and FePc adsorb on an oxygen site of MgO with a (2,1) lattice distance. From DFT we also infer that the Fe atom is located underneath the benzene ring ($C_6H_6$) of the attached FePc ligand (Fig. 1b), thus mimicking the structure found in organometallic arene complexes, in particular half sandwich or piano stool complexes[34]. On surfaces, these kinds of systems have however been only rarely synthesized[35–37]. As a consequence of the shared FePc ligand, the Fe spin under the benzene ring, referred to as $Fe(C_6H_6)$ in the following, is very close to the FePc central spin. In order to probe the joint magnetic properties of the emerging complexes, we first perform d$I$/d$V$ measurements on the $Fe(C_6H_6)$- and FePc-site (Fig. 1c). These reveal a double-step feature at ~20 meV on both sites. We attribute this to inelastic electron tunneling spectroscopy (IETS) excitations from the magnetic ground state to the excited states. Performing spin transport calculations[38], we can reproduce (black lines in Fig. 1c) both the correct position and intensity of the IETS measurements utilizing a Hamiltonian of the form (see Supplementary Section 3).

$$H = J \cdot \vec{S}_{FePc} \cdot \vec{S}_{Fe(C6H6)} + D \cdot S_{z, Fe(C6H6)}^2 \qquad (1)$$

where $J = 14$ meV is the (antiferromagnetic) Heisenberg exchange coupling between the FePc and $Fe(C_6H_6)$ spins and $D = 1.9$ meV is the out-of-plane magnetic anisotropy of $Fe(C_6H_6)$. Interestingly, we find that for $Fe(C_6H_6)$, a spin of $S_{Fe(C6H6)} = 1$ is required: Using $S_{Fe(C6H6)} = 2$ as in the case of the isolated Fe atom, we are not able to reproduce the

experimental data in Fig. 1c (Supplementary Section 3). This reduction of the Fe spin state is additionally supported by remote magnetic sensing experiments (Supplementary Section 4).

To understand the spin state of $Fe(C_6H_6)$ we employ DFT calculations (Details see Supplementary Section 5): In Fig. 1d we show the non-spin-polarized projected density of states (PDOS) of an individual Fe atoms' $s$- and $d$-states as discussed elsewhere[39], leading to four orbitals ($d_{xy}, d_{xz}, d_{yz}$ and $d_{z^2}$) close to half-filling and consequently $S_{Fe} = 2$. To rationalize the spin state of $Fe(C_6H_6)$, we employ a simple model by placing a benzene ring atop the Fe atom (Fig. 1e), as it is the case in the complex and in general for organometallic (half) sandwich complexes[34]. We find that the order of the available states is greatly changed and best described now by those predicted by molecular orbital theory[34]. The molecular orbital diagram of the complex (Supplementary Fig. 7) nicely illustrates, how, in particular, the $e_1$ states of benzene have a strong overlap with the Fe $d_{xz}$ and $d_{yz}$ orbitals and thus contribute to the stability of the complex.

The resulting two frontier orbitals $2e_1$ have strong $d_{xz}/d_{yz}$ character and are half-filled, indicating a spin state of $S_{Fe(C6H6)} = 1$. We note that this resembles to a certain degree the frontier orbitals and spin state found in Nickelocene[40], which consists of two cyclopentadienyl ligands and a Ni ion. We additionally performed DFT calculations of the full FePc-$Fe(C_6H_6)$ that show a similar molecular orbital formation (See Supplementary Section 5). These also support that the remarkably strong exchange coupling of $J \sim 14$ meV, the highest observed for spins on MgO, is mediated via the shared FePc ligand.

Figure 2a illustrates again the energy level diagram of the FePc-$Fe(C_6H_6)$ complex under the action of $D, J$, and external magnetic field $B$ as derived from the spin model in Fig. 1c: For small exchange coupling ($J \sim 0$), the out-of-plane anisotropy $D$ lifts the degeneracies of the $Fe(C_6H_6)$ states. For increasing exchange coupling ($J \neq 0$), two state manifolds form with $S_{tot} \approx \frac{1}{2}$ and $S_{tot} \approx \frac{3}{2}$ which are split by an energy $\frac{3}{2}J$. These transitions between the ground state doublet and the excited states quartet are the transitions observed in the IETS measurements in Fig. 1c and are indicated in Fig. 2a. Figure 2b shows the energy diagram as a function of magnetic quantum number $\langle m_z \rangle$. Since the excited state quartet is further split by $\frac{2}{3}D$, two distinct steps are observed in Fig. 1c. While IETS measurements are well suited to explore the excited state levels, we additionally perform ESR measurements on the complex (Fig. 2c, d) to reveal low lying excitations. Here, a radio frequency (RF) voltage is applied (Fig. 1b) to drive the transition between the two states $|0\rangle$ and $|1\rangle$. Measuring the resonance frequency $f_0$ as a function of external magnetic field $B$ (out-of-plane) permits to determine the magnetic moment $\mu$ of the complex via the resonance condition $hf_0 = 2\mu B$, where $h$ is Planck's constant. We can perform these measurements on the FePc site (Fig. 2c) and the $Fe(C_6H_6)$ site (Fig. 2d) and intriguingly, obtain in both cases $\mu \approx 1\mu_B$, where $\mu_B$ is the Bohr magneton (Supplementary Section 6 and Supplementary Fig. 12). This suggests that the system acts as one magnetic unit, i.e. a strongly coupled spin system with a ground state that involves both spins: Utilizing again the Hamiltonian in Eq. 1, we plot in Fig. 2e the contribution to the ground state wave function of the coupled system: While in the case of $D \gg J$ the ground state $|0\rangle$ is given by $|m_z^{FePc}; m_z^{Fe(C6H6)}\rangle = |\frac{1}{2}; 0\rangle$, for increasing exchange interaction a contribution of $|-\frac{1}{2}; +1\rangle$ is additionally mixed in. In the limiting case of $J \gg D$ the ground state doublet is given as[32]:

$$|0\rangle = \frac{1}{\sqrt{3}} \left| +\frac{1}{2}; 0 \right\rangle - \frac{\sqrt{2}}{\sqrt{3}} \left| -\frac{1}{2}; +1 \right\rangle$$
$$|1\rangle = \frac{\sqrt{2}}{\sqrt{3}} \left| +\frac{1}{2}; -1 \right\rangle - \frac{1}{\sqrt{3}} \left| -\frac{1}{2}; 0 \right\rangle \qquad (2)$$

This leads to some noteworthy consequences: First, besides their more complicated form, the two states form a two-level system with $m_z \approx \pm 1/2$ (See Fig. 2b) which are energetically well separated from

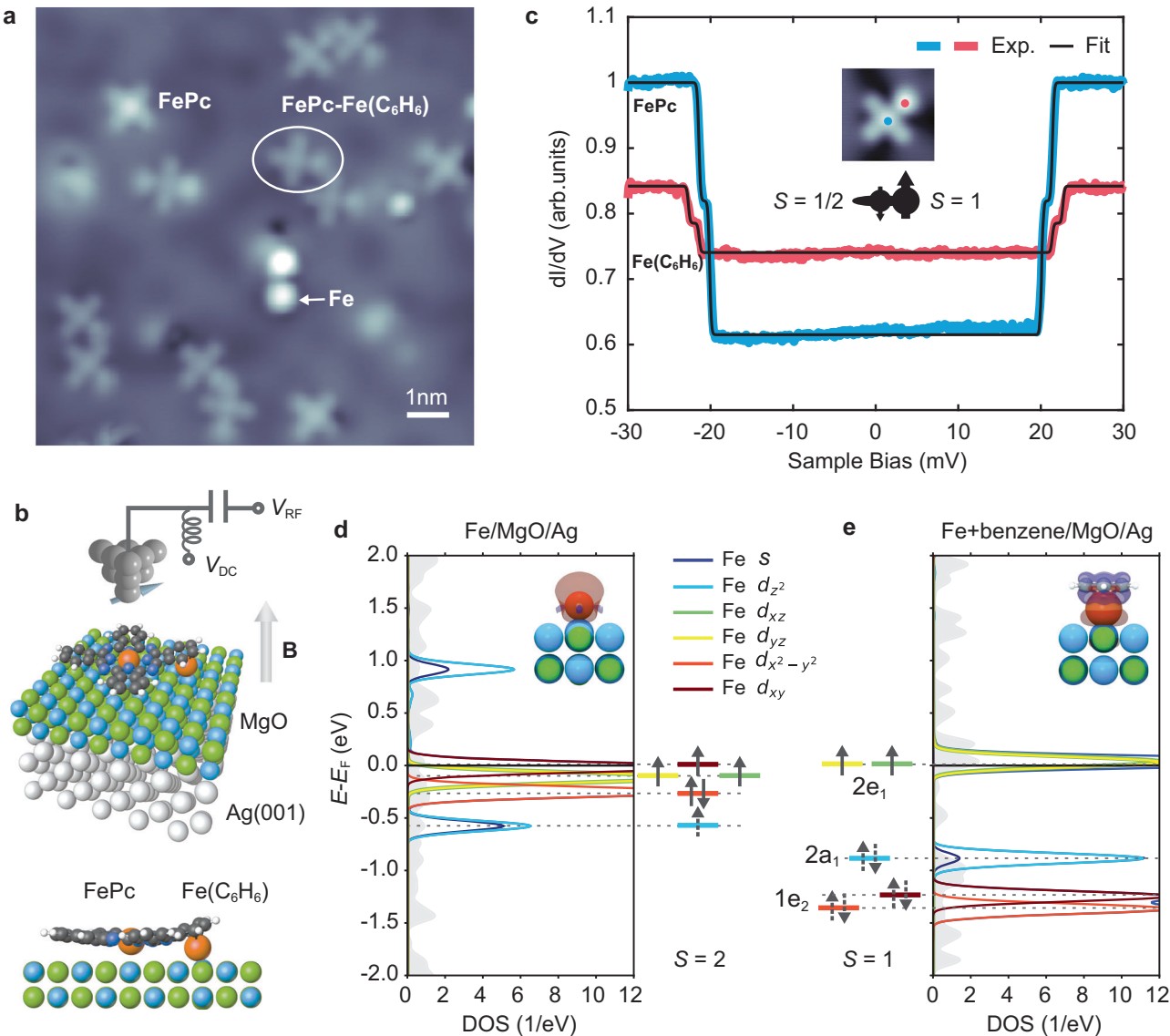

**Fig. 1 | Engineered spin systems in artificial FePc-Fe(C₆H₆) complexes. a** STM Topographic image showing several molecular FePc-Fe(C₆H₆) complexes, along with single FePc molecules, Fe atoms and dimers of the complexes (see also Fig. 4) on 2 ML MgO on Ag(001) ($I = 20$ pA, $V_{DC} = -150$ mV, image size: $10.5 \times 10.5$ nm², $T = 50$ mK). **b** Sketch of the ESR-STM experiment: A radio frequency (RF) voltage $V_{RF}$ is applied to the tunnel junction in addition to the DC bias voltage $V_{DC}$. In addition, a magnetic field **B** is applied out-of-plane. The 3D view and side view image show the stable configuration of the FePc-Fe(C₆H₆) complex calculated by DFT. Orange spheres show the position of the Fe atoms. **c** d$I$/d$V$ spectra acquired at the FePc (blue curve) and Fe(C₆H₆) site (red curve) as indicated in the inset ($I = 100$ pA, $V_{DC} = 30$ mV, $V_{mod} = 0.25$ mV). The double steps are fitted by inelastic electron spin transport calculations (black lines)[38]. The central sketch illustrates the overall spin structure. **d** Projected density of states (PDOS, left) and schematic orbital occupancy (right) for a single Fe atom on 2 ML MgO/Ag(001) obtained from a non-spin-polarized DFT calculation (see inset). It shows the orbitally resolved Fe $s$ and $3d$ orbitals and their Lowdin charges, i.e., projections on atomic states. A strong $4s - 3d_{z^2}$ hybridization (light and dark blue) is found. The dashed arrows indicate shared electrons of partially filled states. **e** PDOS (right) and orbital occupancy (left) of Fe with an added benzene ring atop (see inset). Here, the Fe states shift in energy in the presence of the ring and are additionally labeled by the respective molecular orbital states. The plot shows projections onto the dominating $3d$ states via Lowdin charges, i.e., atomic orbitals (See Supplementary Section 5). The new ligand framework leaves the $2e_1$ orbitals with dominating $d_{xz/yz}$ contribution singly occupied at the Fermi level resulting in S = 1.

the higher energy states. This can be roughly understood from the imbalance of the two spins [e.g., $(1/2 + 0)$ or $(-1/2 + 1)$ for the $|0\rangle$ state], as it is similarly encountered in ferrimagnetism. Second, the two spin states in Eq. (2) are correlated, since they are not separable anymore, as observed similarly for atomic spin systems[41,42]. To quantify this, we use the negativity $\eta$, which is employed as a measure for the pairwise entanglement[41,43] and which becomes non-zero for non-separable states (See Fig. 2 caption and Supplementary section 3). It approaches $\eta \approx 0.3$ for $J \gg D$ (Fig. 2f)[32]. Thus, the correlated non-separable ground state with $m_z \approx \pm 1/2$ explains well why the same $\mu \approx 1\mu_B$ is obtained on both sites in the complex. This is additionally visualized in the spin

contrast map in Fig. 2g: Here, the spin density is delocalized over both Fe(C₆H₆) and FePc sites in the complex.

In order to investigate how the unique spin structure of the ferrimagnet complex affects its coherent spin dynamics, we performed pulsed ESR experiments[27]. Figure 3a illustrates a Rabi oscillation measurement (see Supplementary Section 7) on the Fe(C₆H₆) site [Phase coherence time $T_2^{Rabi} = (48 \pm 9)$ ns]. Figure 3b shows the power-dependent Rabi oscillations, where we observe a linear increase in the Rabi rate $\Omega \propto V_{RF}$ (Supplementary Fig. 13) as expected for an electric field-driven ESR mechanism[25,27]. In the Chevron pattern of the Rabi oscillation (Fig. 3c), the intensity decreases and the oscillation

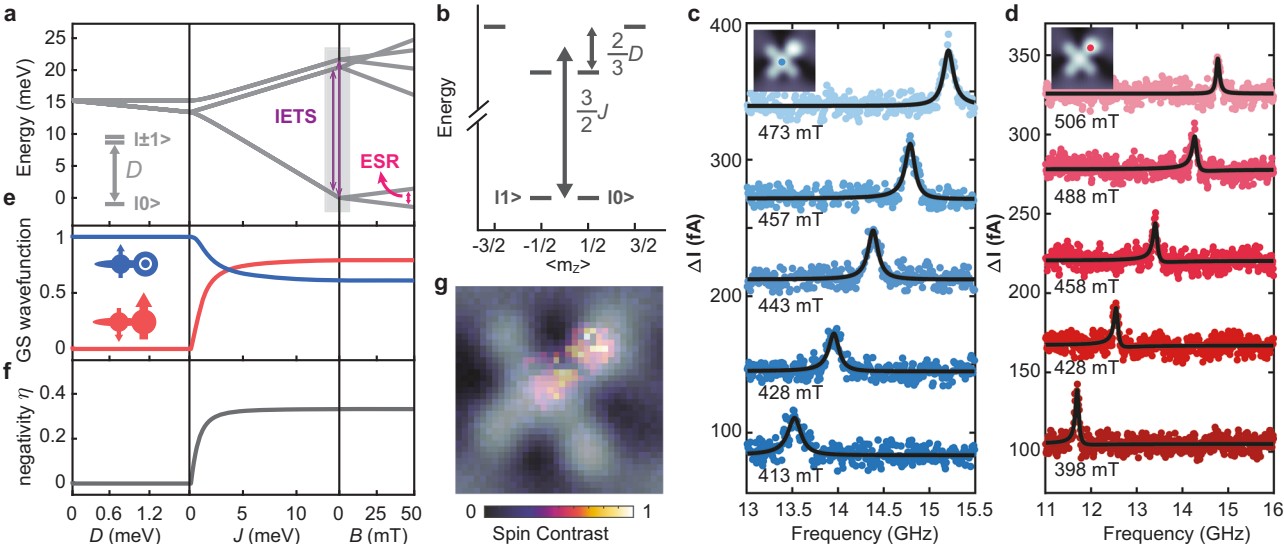

**Fig. 2 | Electron spin resonance (ESR) of a single FePc-Fe(C₆H₆) complex in an STM. a** Energy level diagram resulting from the simulation of a mixed-spin-(1,1/2) Heisenberg ferrimagnet in Eq. 1. The magnetic anisotropy $D$ lifts the degeneracy of a spin 1 at zero field. The exchange coupling ($J$ term) with a spin ½ leads to new $S_{tot} = 1/2$ and $S_{tot} = 3/2$ multiplets, which are both further separated by the Zeeman energy in a magnetic field **B**. **b** Energy level diagram at zero field shows the states as a function of the $m_z$ expectation value, revealing a well-separated ground state doublet. **c** ESR-STM measurements on the FePc site ($V = 70$ mV, $I = 50$ pA, $V_{rf} = 12$ mV) and **d** on the Fe(C₆H₆) site ($V = 25$ mV, $I = 5$ pA, $V_{rf} = 3$ mV) at different magnetic fields. Linear fits to the slope yield a magnetic moment of

$(1.004 \pm 0.012)\,\mu_B$ [FePc site] and $(1.008 \pm 0.007)\,\mu_B$ [Fe(C₆H₆) site]. The contributions to the ground state wave function (**e**) and negativity $\eta = \sum_j \left( \left| \lambda_j \right| - \lambda_j \right)/2$ ($\lambda_j$ being an Eigenvalue) (**f**) as a function of $D$, $J$ and $B$. The cartoon insets in (**e**) illustrate the spin of FePc (left) and Fe(C₆H₆) (right). **g** A spin density map (pseudocolor) with corresponding topography overlaid (gray) of the complex reveals spin contrast on both the FePc and the Fe(C₆H₆) site. The spin signal was mapped by acquiring a spectroscopic map close to zero bias [|d²I/dV²($V = 2$ mV)|] with a spin-polarized tip and normalized to the highest value (1.7 nm × 1.7 nm, setpoints: $V = 25$ mV, $I = 50$ pA, $V_{mod} = 2$ mV).

frequency increases as the system is detuned from the resonance frequency. Ramsey fringe measurements (Fig. 3d) also give a dephasing time ($\sim 19$ ns) of the same order as $T_2^{Rabi}$ and comparable to those found for single on-surface atomic spins[27]. Thus, they are likely limited by the same sources of decoherence, i.e., $V_{RF}$-induced tunneling current and fluctuations in the tip magnetic field[27]. However, we here notably achieve a Rabi frequency of $(95.5 \pm 3.2)$ MHz at $V_{RF} = 60$ mV, which corresponds to a $\pi$-time of $T_\pi = \frac{\pi}{\Omega} = (5.24 \pm 0.17)$ ns. This exceeds the value observed in other spin-1/2 systems, such as Ti[27] and pristine FePc[25], by a factor of ~2 at half the $V_{RF}$ voltage. This can originate from multiple sources[44–46], including a larger displacement of the surface spin, a different coupling to the magnetic tip, as well as the change in energetic positions of the molecular orbitals (Fig. 1e and Supplementary Section 7).

Additionally, we find a strong enhancement of the relaxation time $T_1$ for which we employ an all electrical pump-probe sequence (Fig. 3e)[29,47]: Here, a voltage pulse ($V_{Pump}$) pumps the spin system into an excited state while after a varying delay time $\Delta t$ a second probe pulse ($V_{Probe}$) probes, if the spin still is in the excited state. For direct comparison between FePc-Fe(C₆H₆) and the pristine FePc, we measure on the FePc site for both cases with the same experimental parameters. From exponential-decay fits we obtain $T_1 = 1086$ ns (ferrimagnet complex), and $T_1 = 179$ ns (pristine FePc). By performing setpoint-dependent $T_1$ measurements, we conclude that the lifetime in both cases is limited by inelastic scattering with tunneling electrons from both the metallic substrate and tip[29] [Supplementary Section 8]. Minimizing the latter, we obtain 1.6 μs (ferrimagnet complex) and 0.4 μs (pristine FePc) in the limit of large tip-surface distance (Supplementary Fig. 14). To explain the longer lifetime obtained for the complex, we return to its spin model introduced in Fig. 2: The probability for inelastically scattered tunneling electrons results most crucially from the transfer matrix element $\left| M_{if}^e \right|^2$, where

$M_{if}^e = \left\langle \varphi_f, \psi_f \middle| \frac{1}{2} \mathbf{S} \cdot \boldsymbol{\sigma} + u \middle| \varphi_i \psi_i \right\rangle = m_{if} + u\delta_{if}$[38]. This matrix element connects the initial states in the electron baths $\left| \varphi_i \right\rangle$ and spin system $\left| \psi_i \right\rangle$ to their final states $\left| \varphi_f \right\rangle$ and $\left| \psi_f \right\rangle$. **S** ($\boldsymbol{\sigma}$) is the spin vector operator of the local spin (tunneling electron). Thus, $m_{if}$ contains the inelastic scattering between the two spins, while $u$ describes spin-conserving Coulomb potential scattering. Consequently, the probability for inelastic spin-flip scattering depends on the states $\left| \psi_{i,f} \right\rangle$ of the spin system. Figure 3f shows the evolution of this probability as a function of the exchange coupling $J$ when using the states in Eq. 2 for $\left| \psi_{i,f} \right\rangle$ (Supplementary Section 9): In the limit of small $J$, the system can be assumed to be in the state of pristine FePc for which we obtain ~29%. This drops to ~6% for the complex as the spin system is transitioning into the correlated ground state. This suggests a decrease by ~4–5, which matches well with the results for the pump-probe data. The reduced inelastic scattering can be intuitively understood as tunneling electrons interacting only with the FePc spin, rather than the complete coupled spin system. In addition to the simulation, we can evaluate the suppressed inelastic scattering from spin-polarized IETS d$I$/d$V$ measurements (Supplementary Section 9). The probabilities of inelastic scattering obtained from these measurements are additionally shown in Fig. 3f and agree well with the spin transport calculations.

The results of Fig. 3 demonstrate that the FePc-Fe(C₆H₆) complex allows for coherent manipulation with an enhanced spin lifetime as well as a fast Rabi rate. The latter also benefits from the suppressed inelastic scattering, since this allows us to perform measurements at closer tip-sample distances and thus stronger tip fields without significantly reducing relaxation and coherence. In general, we think that, in particular, the protection makes quantum ferrimagnets a promising system for quantum sensing, simulation, or information processing:

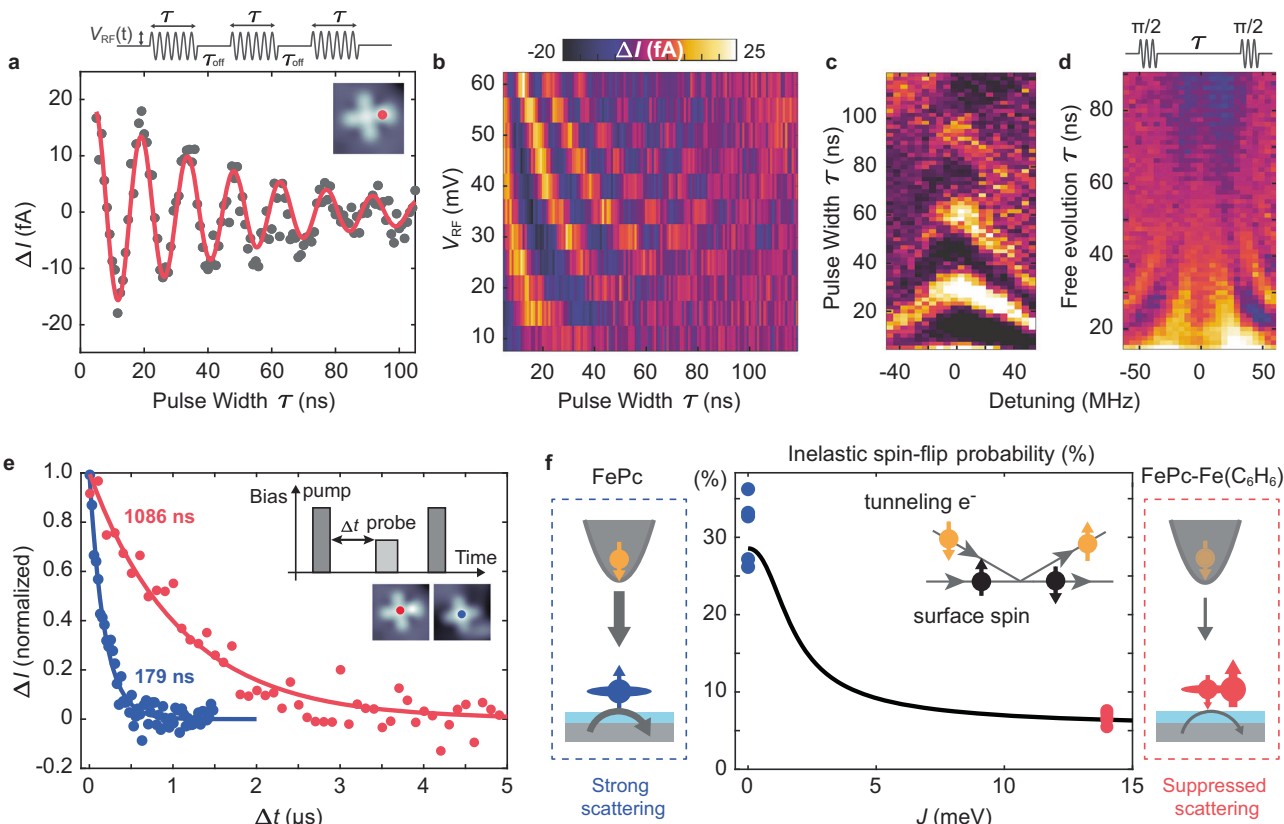

**Fig. 3 | Spin dynamics and coherent control of the complex. a** Rabi oscillation measurement performed at the Fe(C$_6$H$_6$) site in the complex with feedback loop on ($V_{DC}$ = −50 mV, $V_{RF}$ = 45 mV, $I$ = 4 pA, $f_0$ = 14.512 GHz, $B$ = 444 mT, $\tau_{off}$ = 700 ns). Top: RF pulse scheme of width τ and amplitude $V_{RF}$. A constant DC voltage $V_{DC}$ was applied during the sequence as an additional spin initialization and readout following the scheme in ref. 27. The solid red curve is a fit to an exponentially decaying sinusoid: $\Delta I = I_0 \sin(\omega\tau + \alpha) \exp\left(-\tau/T_2^{Rabi}\right)$ for which we obtain $T_2^{Rabi} \sim 48 \pm 9$ ns. **b** Rabi oscillation measurements at different $V_{RF}$. The linear dependence of the Rabi rate on $V_{RF}$ is shown in Supplementary Fig. 13. $T_2^{Rabi} \sim 69 \pm 15$ ns (at $V_{RF}$ = 30 mV). **c** Rabi chevron patterns and **d** Ramsey fringes with varying detuning Δ = ($f - f_0$). [Setpoint in (**c, d**): $V_{DC}$ = −60 mV, $V_{RF}$ = 60 mV, $I$ = 4 pA, $f_0$ = 14.04 GHz, $B$ = 473 mT, $\tau_{cycle}$ = 450 ns in (**c**) and $\tau_{cycle}$ = 350 ns in (**d**). Color scale: [−10 fA ≤ ΔI ≤ 15 fA in (**c**)

and −7 fA ≤ ΔI ≤ 6 fA in (**d**)]. **e** Pump−probe measurements taken on single FePc (blue) and FePc in the complex (red). Solid lines are exponential fits of the form $\Delta I = I_0 \exp(-\tau/T_1)$, resulting in $T_1$ = (1086 ± 188) ns (FePc in complex) and $T_1$ = (179 ± 18) ns (single FePc) [parameters: $I_{set}$ = 25 pA, $V_{set}$ = 200 mV, $B$ = 600 mT, $V_{pump}$ = 80 mV, $V_{probe}$ = 40 mV, $\tau_{pump}$ = 50 ns, $\tau_{probe}$ = 120 ns, $\tau_{cycle}$ = 7.8 µs (FePc in complex), $\tau_{cycle}$ = 1.7 µs (single FePc)]. **f** Inelastic spin-flip probability as a function of $J$ (black line) derived from spin transport calculations[38] and the states in Eq. (2). The experimental inelastic spin-flip probability extracted from dI/dV measurements (see Supplementary Section 9) on five pristine FePc molecules [blue dots, (31.2 ± 1.9)%] and from five complexes [red dots, (6.7 ± 0.3)%] are shown as well. Inset and sketches to the sides illustrate the spin-flip process in which a tunneling electron exchanges momentum with the on-surface spin.

The exchange-like interaction between the complex and the tunneling electrons is similar to sources of relaxation in other quantum architectures, such as flip-flop processes with nuclear spins.

However, a crucial ingredient for creating more complex spin structures is the creation of interacting spin systems from individual building blocks. Figure 4a shows a dimer consisting of two closeby FePc-Fe(C$_6$H$_6$) complexes (several more are shown in Fig. 1). A direct advantage of the complex for spin-spin-coupling is the position of the Fe(C$_6$H$_6$) spin at the FePc ligand. This arrangement facilitates coupling with the adjacent complex due to a shorter distance between spins and leads to much larger J coupling than for dimers of pristine FePc[26]. The coupling scheme is illustrated in Fig. 4b, with the intramolecular coupling $J_1$ = 14 meV as used before and the weaker intermolecular coupling $J_2^{eff}$ (between ferrimagnets). Treating, for now, the two ferrimagnet complexes as effective spin ½ systems leads to a Hamiltonian of the form[26].

$$H = H_1 + H_2 + D_{dip} \cdot \left(\mathbf{S_1} \cdot \mathbf{S_2} - 3\left[\mathbf{S_1} \cdot \hat{\mathbf{r}}\right] \cdot \left[\mathbf{S_2} \cdot \hat{\mathbf{r}}\right]\right) + J_2^{eff} \cdot \mathbf{S_1} \cdot \mathbf{S_2} \quad (3)$$

Where $H_1 = g_1\mu_B\mathbf{S_1} \cdot (\mathbf{B} + \mathbf{B_{tip}})$ marks the single-site Zeeman energy of the left and $H_2 = g_2\mu_B\mathbf{S_2} \cdot \mathbf{B}$ the term of the right complex. Here, $D_{dip}$ is the effective magnetic dipolar coupling between two spins.

To investigate the interacting spin system, we perform ESR measurements on one of the spins for varying tip magnetic field $B_{tip}$. Since $B_{tip}$ contributes in a Zeeman-like way, but only to one of the complexes[26], it allows to tune the coupled spin system through an avoided level crossing as shown in Fig. 4c. Here, we depict the energy level diagram of two coupled spin-½ as a function of $B$ and $B_{tip}$. We additionally highlight the dominating ESR transitions below ($f_3$ and $f_4$) and above ($f_1$ and $f_2$) the point of no detuning δ = $g_1\mu_B(B + B_{tip}) - g_2\mu_B B = 0$.

At the latter, the single-site Zeeman energies for the complex on the left ($H_1$) and right ($H_2$) are equal. These kinds of measurements have been realized before on regular coupled spin ½ systems, i.e., Ti atoms[48,49] and pristine FePc[26]. We show in the following that also the more complex ferrimagnetic spins can be coupled as well and in fact constitute more flexible magnetic building blocks. An ESR spectrum measured on the Fe(C$_6$H$_6$) site within the left complex reveals four ESR peaks (Fig. 4d), corresponding to those labeled in Fig. 4c. Figure 4e shows an ESR map measurement as a function of frequency $f$ and tip setpoint current $I$. The latter is in good approximation ∝ $B_{tip}$[46] and can tune the frequency and intensity of the four ESR transitions. The evolution shows good agreement with the predictions of an

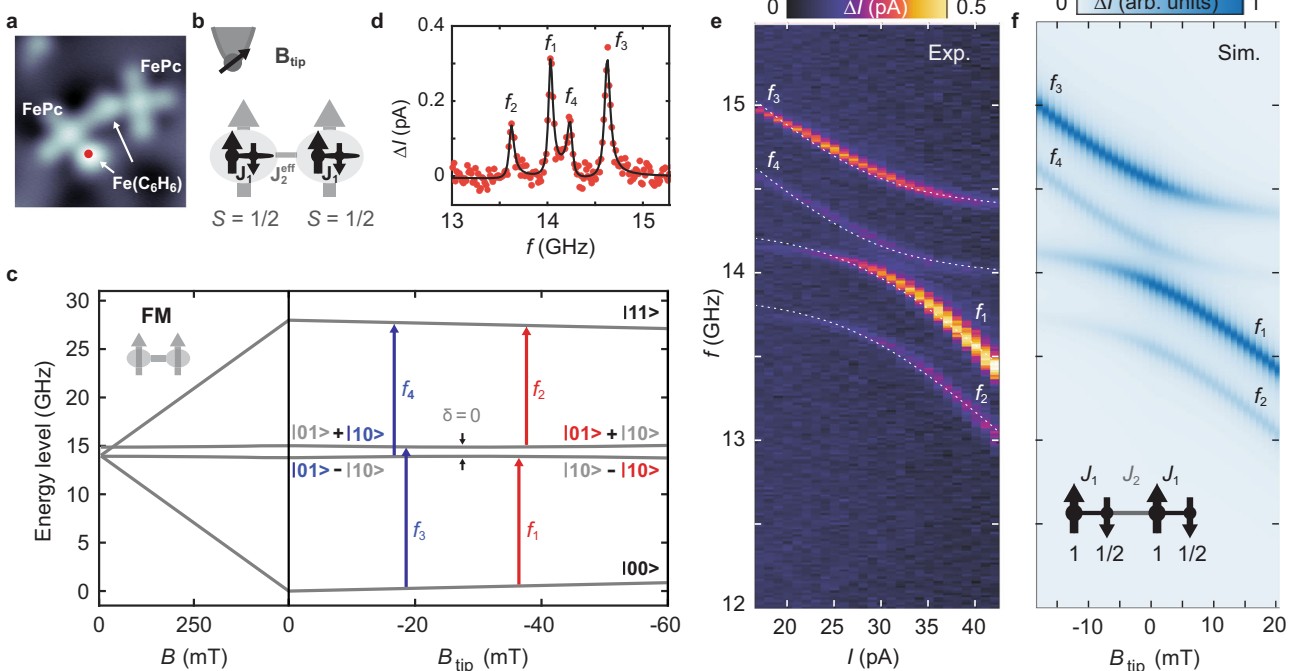

**Fig. 4 | Spin coupling in a dimer of ferrimagnet complexes. a** Topographic image of two coupled complexes ($I = 10$ pA, $V_{DC} = 100$ mV, image size: $3 \times 3$ nm²) with a spacing of 1.04 nm [(3,2) MgO lattice sites]. **b** Illustration of the magnetic coupling within the spin structure: The small (large) black arrows represent the FePc [Fe(C₆H₆)] spins. Within each individual complex two spins are antiferromagnetically coupled with strength $J_1$. Since also the FePc and Fe(C₆H₆) in different complexes couple antiferromagnetically, the overall coupling between the complexes is ferromagnetic with coupling strength $J_2^{eff}$. **c** Energy level diagram of two coupled spin ½ in the presence of $B$ and $B_{tip}$. The detuning $\delta$ quantifies the distance from the avoided level crossing. The blue (red) arrows represent the dominant ESR transitions at weak (strong) tip fields. **d** ESR frequency sweep

measurement at a fixed $B_{tip}$ showing four peaks $f_1$ to $f_4$, corresponding to the transitions labeled in (**c**). All data are measured on the Fe(C₆H₆) site position marked as a red dot in (**a**). ($V_{DC} = 60$ mV, $I_{set} = 29$ pA, $V_{RF} = 12$ mV, $B = 473$ mT). The black line is a fit of four Lorentzians to the data. **e** ESR frequency sweep measurements at different setpoint currents ($B_{tip}$) showing an avoided level crossing ($V_{DC} = 60$ mV, $V_{RF} = 12$ mV, $B = 473$ mT). The dotted white curves represent fitted ESR transitions using a two-spin model, with $J_2^{eff} = -531$ MHz and $D_{dip} = 131$ MHz, $g_1 = 2$, $g_2 = 2.01$. **f** Simulation of the coupled spin system using a four-spin model (inset) of spins (1, 1/2, 1, 1/2) with antiferromagnetic coupling strengths $J_1$ and $J_2$. The fitting parameters are: $J_1 = 14.65$ meV, $J_2 = 1.2$ GHz, $g_{FePc} = 2$, $g_{Fe} = 2$.

effective two-spin model (Fig. 4e, see Supplementary Section 10) which suggests that the two complexes can be described as two coupled effective spin ½ with intermolecular coupling strength $J_2^{eff}$. This makes them suitable building blocks for larger spin systems. However, a notable difference compared to previous works on atomic and molecular spins is that $J_2^{eff}$ is negative, constituting a ferromagnetic (FM) exchange interaction between the two effective spins. This can be rationalized when considering the internal structure of each complex (Fig. 4b). Here, the coupling between all adjacent spins remains antiferromagnetic, as it is most frequently found for superexchange interaction. However, the two ferrimagnet complexes can also be coupled antiferromagnetically (AFM, $J_2^{eff} > 0$) when positioning the two Fe(C₆H₆) centers in each complex closest together (Supplementary Figs. 18, 19 and Supplementary Section 10).

To capture minute details in the position and intensity of the peaks (see Supplementary Section 10), we employ a simulation of the full four-spin system (Fig. 4f), where all magnetic dipolar couplings are calculated based on the distances between all spin centers (Supplementary Fig. 20). The full four-spin model does not show significant deviations from the two-spin model for both the FM case in Fig.4f and the AFM case discussed in Supplementary Fig. 19. Only the exchange coupling differs for the effective two-spin model compared to a four-spin model, since it now incorporates the different spin structure of the complexes. Thus, we conclude that the complexes can be effectively treated as spin ½ systems and that they can serve as fundamental building blocks for larger spin structures.

## Discussion

Our measurements highlight how both atomic and molecular building blocks can be combined to tailor new molecular spin structures. A crucial ingredient is the bond formation between the Fe and the benzene ring, which, instead of using tip-assisted on-surface assembly, could also be created by a thermally driven on-surface self-assembly. This was already successfully demonstrated on metal substrates[50,51] by employing thermal sample annealing. We also expect that these systems could be directly synthesized in fully chemically derived molecular ferrimagnets.

In particular, the intricate connection between the emerging correlated spin state of the complex and the improvement in its dynamic properties reveals a key strategy to protect spins on surfaces from the main source of relaxation, i.e., scattering by nearby electron spins. Combining this protection with an increased MgO layer thickness[29] and a remote spin readout[28] could allow to further enhance $T_1$ and $T_2$. For a remote spin readout, the demonstration of efficient spin-spin coupling (Fig. 4) is an important prerequisite, which also offers the opportunity to design larger spin structures or quantum simulators with both AFM and FM coupling. Lastly, we note that the concept of strongly coupled quantum ferrimagnets employed here is not unique to assembled molecular spins on surfaces: The ingredients for the spin system are quite universal and could also be realized by spins in other quantum architectures, including quantum dots, color centers or fully chemically derived molecular ferrimagnets.

## Methods

### Sample preparation

All sample preparation was carried out in situ at a base pressure of $<5 \times 10^{-10}$ mbar. The Ag(001) surface was prepared through several cycles of argon-ion sputtering and annealing through e-beam heating. For MgO growth, the sample was heated up to 510 °C and exposed to an Mg flux for 10 min in an oxygen environment at $10^{-6}$ mbar, leading to an MgO coverage of ~50% and layer thicknesses ranging from two to five monolayers. FePc was evaporated onto the sample held at room temperature using a home-built Knudsen cell at a pressure of $9 \times 10^{-10}$ mbar for 90 s. Electron-beam evaporation of Fe was carried out for 21 s onto the cold sample. We determined the thickness of MgO layers through point-contact measurements on single Fe atoms[29]. All experiments were carried out using a Unisoku USM1600 STM inside a home-built dilution refrigerator with a base temperature of 50 mK. An effective spin temperature of ~300 mK was estimated from ESR measurements of Fe dimers. Here, the intensities of ESR peaks of the coupled electron spin states depend on temperature[52], which we take as an estimate of the Boltzmann distribution in the experiment.

### Pulsed electron spin resonance measurements

Spin-polarized tips were prepared by picking up individual Fe atoms from the surface. The spin polarization was tested through the asymmetry in d$I$/d$V$ spectra of FePc with respect to voltage polarity. Magnetic tips showing a high spin contrast were subsequently tested in continuous-wave (CW) ESR-STM measurements. The RF voltage was applied to the tip side of the junction using an RF generator (Rohde & Schwarz SMB100B). The RF voltage was combined with the DC tunnel bias using a Diplexer (Marki Microwave MDPX-0305). Note that while the bias voltage was applied to the STM tip, all bias signs were inverted in the manuscript to follow the conventional definition of bias voltage with respect to the sample bias.

For pulsed ESR measurements, we followed the manipulation scheme introduced in ref. 27. An arbitrary waveform generator (Zurich Instruments HDAWG) gated the output of the RF generator to generate the desired pulsed ESR scheme. We used a digital lock-in amplifier (Stanford Research Systems SR860) to read out the ESR signal using an on/off modulation scheme at 323 Hz. During the Rabi or Ramsey measurements, the feedback loop was set to low-gain values. In total, these measurements took ~3 s per data point, so that a typical Rabi or Ramsey measurement, including three iterations of 100 points, took around 15 min. All measurements were conducted by using schemes described in ref. 27, in which a DC background current is applied in lock-in A and B cycles, while RF pulses were only applied in lock-in A cycles. Linear backgrounds that originate from an additional rectification for increasing pulse lengths[27] have been subtracted.

For pump-probe measurements in Fig. 3 and Supplementary Fig. 14, we altered the tunnel current by changing the bias voltage to zero once the feedback loop had been disengaged. Consequently, all measurements have been performed at constant tunnel conductance and constant tip-atom distance, which in particular helped to keep the influence of the tip field constant. For each measurement, the lockin A cycles include both pump and probe pulse voltages while the lock-in B cycles are left empty, with no DC voltages applied.

### DFT calculations

All periodic-cell density functional theory (DFT) calculations were performed using plane-wave basis and pseudopotentials in Quantum Espresso version 7.1[53,54]. All pseudopotentials use the Perdew–Burke–Ernzerhof (PBE) parametrization for the exchange and correlation potential[55]. We mimic the experiment by placing the Fe on top of MgO (100), which is supported by a silver (100) substrate. The bulk lattice constants for silver and MgO with PBE are $a_{Ag} = 4.16$ Å and $a_{MgO} = 4.25$ Å, which results in a lattice mismatch of about 2% (experimental value: 2.9%[55]). To construct the surface slab, we used four layers

of Ag with the lateral lattice constant fixed to that of the PBE bulk silver and added up to two layers of MgO. We created lateral supercells of about $20 \times 20$ Å and 15 Å of vacuum were used to pad the cell in z-direction. All calculations used a k-grid equivalent to $15 \times 15 \times 1$ k-points of the $1 \times 1 \times 1$ unit cell. All 3d plots in the main text were created using OVITO ref. 56. To account for dispersive forces we used the revised VV10 functional in all calculations[57]. We used non-spin-polarized (similar to open-shell restricted Hartree-Fock) calculations for the calculation of the crystal-field splittings and molecular orbital order, as well as spin-polarized (similar to unrestricted Hartree-Fock) calculations to confirm the spin-states of the adsorbates. The calculated STM topographic images (Supplementary Fig. 22) use the Tersoff-Hamann approach as implemented in the STMpw code[58]. They show generally good agreement with respect to the individual features of Fe and FePc in the dimer at the respective biases.

## Data availability

The data supporting the findings of this study are available in the article. Source data are provided with this paper.

## Code availability

The code for the DFT calculated STM images is available from Zenodo[58] and the code for DFT calculations is available from Quantum Espresso page (https://www.quantum-espresso.org/).

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

## Acknowledgements

P.W. acknowledges funding from the Emmy Noether Program of the DFG (WI5486/1-1), financing from the Baden-Württemberg Foundation Program on Quantum Technologies (Project AModiQuS) and support from the Centre for Integrated Quantum Science and Technology (IQST). P.G. and P.W. acknowledge financial support from the Hector Fellow Academy (Grant No. 700001123). C.W. and J.R. acknowledge support from the Institute for Basic Science (IBS-R027-D1). The authors acknowledge the use of Blender for rendering the images in Fig. 1b and Supplementary Fig. 2b.

## Author contributions

P.W. and W.H. conceived the research. W.H., M.S., P.G., K.H.A.-Y., D.S., C.S., W.W. and P.W. set up the experiment and conducted the measurements. W.H., M.S., P.G., K.H.A.-Y. and P.W. analyzed the experimental data. C.W. performed the DFT calculations. J.R. participated in the modeling of the spin systems. W.H., M.S., P.G., K.H.A.-Y., C.S., W.W. and P.W. discussed the results. W.H. and P.W. wrote the manuscript with input from all authors. W.W. and P.W. supervised the project.

## Funding

## Competing interests

The authors declare no competing interests.
