## [Peer Review File · Nature Communications]

Quantum Spin-Engineering in On-Surface Molecular Ferrimagnets

Corresponding Author: Dr Philip Willke

Version 0:

Reviewer comments:

Reviewer #1

(Remarks to the Author)

The manuscript by Willke et al. is a spectacular combined experimental and theoretical investigation of the coherence properties of a molecular magnetic dimer adsorbed on MgO surface. Their systems is made by a FePc molecule which interacts with a Fe adatom through one of its phenyl rings, and it has been characterized by STM, IETS and ESR-STM. All experimental data are supported by a profound DFT investigation from the single adatom to the entire dimeric system. The manipulation of multi-spin architectures at the single molecule level by STM-ESR is a crucial step towards the addressability and scalability of quantum computing platforms based on molecular spins. At the same time, the synergy between state-of-the-art techniques and DFT simulations is mandatory to shed light on the electronic structure of adsorbed open-shell molecules. This combined effort has been carried out here at a very high quality. Finally, the increase of T1 relaxation time is an important finding that indicates how the exchange interaction can protect the QuBit from decoherence: a similar phenomenon has already been observed for Single Molecule Magnets and it led to striking advances in the field. For all the reasons above, I believe that the manuscript deserves to be published in Nature Communications after addressing a few minor issues.

- The simulation of the STM images should be attempted employing DFT, to compare them with the experimental ones at different biases.
- The authors did not use any dispersion correction (e.g. rVV10) in their DFT optimization. However, they should not be neglected when modeling physisorbed molecules, such as FePc. Is there any evidence that for this class of systems, dispersion corrections are not necessary?
- What is the adsorption energy of the FePc system on Fe/MgO?
- The orbital's energies and projected DOS of Figures 1d, 1e, S8, and S9 show no spin polarization. Did the authors report only the spin-up orbitals? In such a case, also the spin-down channel should be shown at least in SI.
- I found very clear the explanation about the spin state of the Fe-C6H6 system, Roald Hoffmann would be very pleased. However, I have some doubts about the spin multiplicity of the pristine iron on MgO. Why should it present an S=2? I understand that there is an electron that is shared between a dz2 and an s orbital, but did you try to compute the energy of different spin multiplicities with the same protocol employed in the manuscript?
- Some further details about the 'negativity' quantity should be provided in the text.
- The whole part regarding the 'probability for inelastically scattered tunneling electrons' is a bit obscure for a general audience. I believe that some further explanation of the physical meaning of the equation on page 9 should be provided. Which is the physical reason why the inelastic scattering probability is reduced in the dimer? I understand the mathematical equations below, which are also described in the ESI section, but it is not evident from equation 15 where the exchange is playing a role in reducing the transfer matrix element. Moreover, from equation 15 how is the dI/dV quantity computed (the reported data in Figures S14 and S15)?
- Regarding the dimer of complexes, did you try to measure the ESR on different parts of the molecule? Do you expect any variation in the ESR?
- The cartesian coordinates of all the simulated models should be attached to the manuscript.

Reviewer #2

(Remarks to the Author)

Huang et al. demonstrate the engineering of ferrimagnetic dimer complexes presenting strongly coupled spins and improved spin lifetimes. They also demonstrated how these dimers can be used to build ferromagnetic and antiferromagnetic structures. They present high quality data combined with a theoretical analysis that supports and explains properly their experimental achievements. I found the results appealing and suitable to publication on Nature Communications. I have only minor comments:

1. The colors used in the DFT models (figures 1b and S2b) should be changed. The bright colors of the substrate's atoms make it hard to visualize the carbon atoms.
2. In lines 198, 252 and 326 the authors start statements with "We believe", which sounds unscientific. These statements should be removed from the text or rewritten with a more scientific argumentation.
3. In line 315 the authors mention some simulations that are presented in the SI, a short description of the results should also be added in the main text.
4. In line 322 the authors mention that the bond could be formed by on-surface chemistry techniques. Have the authors tried this approach? The tip-induced assembly is limited, and this alternative method would increase the impact of the work.

Version 1:

Reviewer comments:

Reviewer #1

(Remarks to the Author)

The authors addressed all my comments in detail. Only one last issue should be addressed before publication. In their reply, the authors refer to a 'non-polarized calculation' to extract the crystal field. This should be explicitly cited in the figure caption, and the details of this calculation should be added to the computational methods. Is it a restricted open-shell DFT simulation?

Reviewer #2

(Remarks to the Author)

The authors addressed all my comments. I recommend the manuscript for publication.

Reviewer #1 (Remarks to the Author):

The manuscript by Willke et al. is a spectacular combined experimental and theoretical investigation of the coherence properties of a molecular magnetic dimer adsorbed on MgO surface. Their system is made by a FePc molecule which interacts with a Fe adatom through one of its phenyl rings, and it has been characterized by STM, IETS and ESR-STM. All experimental data are supported by a profound DFT investigation from the single adatom to the entire dimeric system.

The manipulation of multi-spin architectures at the single molecule level by STM-ESR is a crucial step towards the addressability and scalability of quantum computing platforms based on molecular spins. At the same time, the synergy between state-of-the-art techniques and DFT simulations is mandatory to shed light on the electronic structure of adsorbed open-shell molecules. This combined effort has been carried out here at a very high quality. Finally, the increase of T1 relaxation time is an important finding that indicates how the exchange interaction can protect the Qubit from decoherence: a similar phenomenon has already been observed for Single Molecule Magnets and it led to striking advances in the field. For all the reasons above, I believe that the manuscript deserves to be published in Nature Communications after addressing a few minor issues.

We thank the reviewer for the support and appreciate the additional comments, which we believe have improved the manuscript. We address them below.

-The simulation of the STM images should be attempted employing DFT, to compare them with the experimental ones at different biases.

Reply: Thank you for the suggestion. We have now included a comparison between DFT-simulated and experimental STM images in the supplementary information. We find good agreement between the experimental topographic images and the calculated ones: At negative bias voltages, we observe a strong feature localized on the Fe(C₆H₆), associated with the states shown in Fig. 1e in the main text. In contrast, the highest occupied molecular orbitals (HOMO) of FePc only appear at around -2 V. At positive bias voltages, we mostly observe the lowest unoccupied molecular orbital (LUMO) of FePc. The measured bias range is limited by the stability of the complex, which destabilizes at higher voltages.

Fig.SM. Comparison of the experimental and DFT calculated topographic images of FePc-Fe(C₆H₆) on MgO/Ag. (a) Large-range dI/dV spectrum at the FePc center, Fe(C₆H₆) site (setpoint: V = 1.5 V, I = 5 pA). (b) Upper panel: the DFT-calculated STM images at different biases calculated in the Tersoff-Hamann approximation. Lower panel: the experimental STM images measured at different biases (constant-current mode, I = 5 pA) corresponding to HOMO, in-gap and LUMO energy, respectively.

-The authors did not use any dispersion correction (e.g. rVV10) in their DFT optimization. However, they should not be neglected when modeling physisorbed molecules, such as FePc. Is there any evidence that for this class of systems, dispersion corrections are not necessary?
 Reply: We apologize for the confusion in the DFT methods section. All our calculations indeed use rVV10 dispersion correction, which in our experience gives the best results for a mixed metal-insulator-molecule system. We have amended the methods section with this information.

“To account for dispersive forces we used the revised VV10 functional in all calculations. [R. Sabatini et al., PhysRevB.87.041108]”

-What is the adsorption energy of the FePc system on Fe/MgO?

Reply: We calculated the adsorption energy of FePc on Fe/MgO/Ag by comparing the total energies of each system without further relaxation of the individual components FePc and the Fe/MgO/Ag substrate.

$$E_{ads} = E_{system} - (E_{FePc} + E_{substrate}) = 3.17 \text{ eV}$$

We note that this energy is most likely not the limiting energy scale for the mechanical stability of the atom-molecule complex as the individual components tend to diffuse at much lower energies.

We added these sentences to the general description of the DFT in the methods section.

-The orbital's energies and projected DOS of Figures 1d, 1e, S8, and S9 show no spin polarization. Did the authors report only the spin-up orbitals? In such a case, also the spin-down channel should be shown at least in SI.

Reply: The observation is correct. However, the DOS plotted is not a single channel from a spin-polarized calculation but rather a non-polarized one. This is due to the fact that the crystal field extracted from this calculation is then used in a spin-model and the spin properties for the system are added in the second step of this procedure. We have amended the supporting information with spin-polarized DOS plots for completeness.

Fig.SM. Projected density of states (PDOS) for Fe/MgO/Ag (left) and Fe+benzene/MgO/Ag (right). The contributions of Fe:d (blue), Fe:s (red), and C:p (green) orbitals are shown. The energy axis is referenced to the Fermi level ($E - E_F = 0$). The spin-resolved DOS is depicted, with positive values corresponding to spin-up (n_\uparrow) and negative values to spin-down (n_\downarrow). The inclusion of benzene modifies the electronic structure, introducing additional contributions from C:p states and shifting the Fe:d-state distribution.

-I found very clear the explanation about the spin state of the Fe-C6H6 system, Roald Hoffmann would be very pleased. However, I have some doubts about the spin multiplicity of the pristine iron on MgO. Why should it present an S=2? I understand that there is an electron that is shared between a dz2 and an s orbital, but did you try to compute the energy of different spin multiplicities with the same protocol employed in the manuscript?

Reply: We thank the reviewer for the kind words. Concerning the Fe atom: The spin state of Fe on MgO when adsorbed on the top site has been first studied by STM spectroscopy and X-ray magnetic circular dichroism (XMCD), and later by ESR-STM and has consistently been found to be S=2 with a significant contribution of the orbital moment to the total magnetic moment (e.g. Baumann *et al.*, Phys. Rev. Lett. 115, 237202 (2015), Choi *et al.* Nature Nano 12, (2017)). From the theory point of view this also required some detailed magnetic multiplet calculations: One of us has studied the reliability of “plain” DFT (Wolf *et al.*, JPCA 124, 11, 2020). In brief, DFT fails to accurately capture excitation barriers for this system due to a lack of accuracy in part of the orbital contribution to the magnetism. Nevertheless, we computed the excited states of Fe on MgO as given below:

S	0	1	2
Total energy (Ry)	-7992.16170521	-7992.19894709	-7992.23902285
Delta E (meV)	1051.5	545.03	0

Table 1: excitation energies for Fe on MgO based on total energies from constrained DFT calculations for S=0, 1, 2.

-Some further details about the ‘negativity’ quantity should be provided in the text.

Reply: A short definition was already given in the figure caption. We have now referenced this more clearly and added a detailed discussion in the supplementary information section 3.

“To explore the entanglement of the mixed spin-(1/2, 1) Heisenberg ferrimagnet, we employ the negativity serving as a measure of the pairwise entanglement^{3,7,8}

$$\eta = \sum_j (|\lambda_j| - \lambda_j)/2 \quad (15)$$

which is defined through eigenvalues λ_j of a partially transposed density matrix $\rho^{T_{1/2}}$ where ρ denotes the full density matrix. Here, $T_{1/2}$ denotes a partial transposition with respect to one subsystem (i.e. the spin-1/2).

Using the respective density operator $\rho = (|0\rangle\langle 0| + |1\rangle\langle 1|)/2$, one gets the following zero-temperature value of the negativity at zero magnetic field³:

$$\eta = \frac{\sqrt{\left(1 - 2\frac{D}{J}\right)^2 + 8} - \left(1 - 2\frac{D}{J}\right)}{4\sqrt{\left(1 - 2\frac{D}{J}\right)^2 + 8}} \times \left[\frac{5\sqrt{\left(1 - 2\frac{D}{J}\right)^2 + 8} + 3\left(1 - 2\frac{D}{J}\right)}{\sqrt{\left(1 - 2\frac{D}{J}\right)^2 + 8} - \left(1 - 2\frac{D}{J}\right)} - 1 \right] \quad (16)$$

For $J \gg D$, $\eta = \frac{1}{3}$.

-The whole part regarding the ‘probability for inelastically scattered tunneling electrons’ is a bit obscure for a general audience. I believe that some further explanation of the physical meaning of the equation on page 9 should be provided. Which is the physical reason why the inelastic scattering probability is reduced in the dimer? I understand the mathematical equations below, which are also described in the ESI section, but it is not evident from equation 15 where the exchange is playing a role in reducing the transfer matrix element. Moreover, from equation 15 how is the dI/dV quantity computed (the reported data in Figures S14 and S15)?

Reply: We agree with the reviewer that this is a bit hard to follow in the current version. Following the suggestion, we now amended the supplementary section by the detailed calculation of the transfer matrix elements:

“While the continuous evolution is obtained numerically, in the limiting case of large J (maximum entanglement) and $J = 0$ (pristine FePc), the ratio can also be calculated directly: The transfer matrix element is determined by the initial and final states and the spin operators:

$$\hat{\sigma} \cdot \hat{S} = \frac{1}{2}\hat{\sigma}_+\hat{S}_- + \frac{1}{2}\hat{\sigma}_-\hat{S}_+ + \hat{\sigma}_z\hat{S}_z,$$

which governs spin-exchange processes between tunneling electrons and the surface spin (Supplementary Fig. 15). In the FePc-Fe(C₆H₆) complex, the exchange interaction modifies the initial and final states, ultimately changing the transfer matrix elements as summarized in the table below. Since the U parameter remains the same for pristine FePc and FePc in the complex, the elastic scattering caused by potential scattering U is the same.

Pristine FePc	FePc in the complex	Fe(C ₆ H ₆) in the complex
Ground state: $ 0\rangle = \left +\frac{1}{2}\right\rangle$, $ 1\rangle = \left -\frac{1}{2}\right\rangle$	Ground state: $ 0\rangle = \frac{1}{\sqrt{3}}\left +\frac{1}{2}; 0\right\rangle - \frac{\sqrt{2}}{\sqrt{3}}\left -\frac{1}{2}; +1\right\rangle$, $ 1\rangle = \frac{\sqrt{2}}{\sqrt{3}}\left +\frac{1}{2}; -1\right\rangle - \frac{1}{\sqrt{3}}\left -\frac{1}{2}; 0\right\rangle$	
$\langle 0 \hat{S}_+ 1\rangle = 1$	$\langle 0 \hat{S}_+ 1\rangle = -1/3$	$\langle 0 \hat{S}_+ 1\rangle = 4/3$
$\langle 1 \hat{S}_- 0\rangle = 1$	$\langle 1 \hat{S}_- 0\rangle = -1/3$	$\langle 1 \hat{S}_- 0\rangle = 4/3$
$\langle 0 \hat{S}_z 0\rangle = 1/2$	$\langle 0 \hat{S}_z 0\rangle = -1/6$	$\langle 0 \hat{S}_z 0\rangle = -2/3$
$\langle 1 \hat{S}_z 1\rangle = -1/2$	$\langle 1 \hat{S}_z 1\rangle = 1/6$	$\langle 1 \hat{S}_z 1\rangle = 2/3$
$U = 1/2$	$U = 1/2$	$U = 1$

Following Ref. [Ternes, New Journal of Physics 17, 063016 (2015).], we can rewrite the spin-exchange scattering term in equation (19) as

$$\begin{aligned}
 |m_{if}|^2 &= \left| \langle \varphi_f, \psi_f | \frac{1}{2} \mathbf{S} \cdot \boldsymbol{\sigma} | \varphi_i, \psi_i \rangle \right|^2 \\
 &= \frac{1}{4} \left(\frac{1}{2} |\langle \varphi_f, \psi_f | \hat{S}_- | \varphi_i, \psi_i \rangle|^2 + \frac{1}{2} |\langle \varphi_f, \psi_f | \hat{S}_+ | \varphi_i, \psi_i \rangle|^2 + |\langle \varphi_f, \psi_f | \hat{S}_z | \varphi_i, \psi_i \rangle|^2 \right)
 \end{aligned}$$

The probability for inelastically scattered tunneling electrons is then given by the ratio of spin-flip processes and all tunneling channels (Supplementary Fig. 15). This results in:

$$P = \frac{\frac{1}{2} |\langle 1|\hat{S}_-|0\rangle|^2}{\frac{1}{2} |\langle 1|\hat{S}_-|0\rangle|^2 + |\langle 0|\hat{S}_z|0\rangle|^2 + 4|\langle 0|U|0\rangle|^2}$$

Which could equivalently be formulated for $|\langle 0|\hat{S}_+|1\rangle|^2$. Using this expression, we obtain P = 28.6% for pristine FePc, 16.7% for Fe(C₆H₆) in the complex, and 5.1% for FePc in the complex, as shown in Supplementary Fig. 16a . This demonstrates the effect of exchange interaction in the complex on inelastic scattering.

-Regarding the dimer of complexes, did you try to measure the ESR on different parts of the molecule? Do you expect any variation in the ESR?

Reply: We conducted the same ESR measurements on the Fe(C₆H₆) site of the other complex and observed similar behavior. For clarity, we present a direct comparison below. The left

panel shows data measured on the left complex, the same as in Fig. 4, while the right panel displays data measured on the right complex using the same tip and measurement parameters. Both exhibit avoided level crossing behavior, with the difference being a frequency shift attributed to variations in the tip magnetic field B_{tip} . Additionally, we attempted measurements on the FePc center; however, detecting ESR signals at FePc center requires a very large B_{tip} , which pushes the system out of the avoided level crossing regime, making the measurements challenging. We added this figure to the supplementary information.

Fig.SM. ESR frequency sweep measurements at different setpoint currents (B_{tip}) showing an avoided level crossing in both complexes within a dimer. ($V_{DC} = -60$ mV, $V_{RF} = 12$ mV, $B = 484$ mT).

-The cartesian coordinates of all the simulated models should be attached to the manuscript.

Reply: We agree with the reviewer. We will upload the cartesian coordinates as supplemental data to the manuscript.

Reviewer #2 (Remarks to the Author):

Huang et al. demonstrate the engineering of ferrimagnetic dimer complexes presenting strongly coupled spins and improved spin lifetimes. They also demonstrated how these dimers can be used to build ferromagnetic and antiferromagnetic structures. They present high quality data combined with a theoretical analysis that supports and explains properly their experimental achievements. I found the results appealing and suitable to publication on Nature Communications. I have only minor comments:

We thank the reviewer for the feedback and the kind words. We address the minor comments below.

1. The colors used in the DFT models (figures 1b and S2b) should be changed. The bright colors of the substrate's atoms make it hard to visualize the carbon atoms.

Reply: We agree with the reviewer that this should be improved. We have improved the visibility by changing the colors of carbon atoms (See below).

2. In lines 198, 252 and 326 the authors start statements with “We believe”, which sounds unscientific. These statements should be removed from the text or rewritten with a more scientific argumentation.

Reply: We have revised these sentences and removed the ‘We believe’. All sentences have been slightly altered to ensure a more scientific argumentation.

3. In line 315 the authors mention some simulations that are presented in the SI, a short description of the results should also be added in the main text.

Reply: We have added a description of the results in the main text: ‘The full four-spin model does not show significant deviations from the two-spin model for both the FM case in Fig.4f and the AFM case discussed in Fig. S19. Only the exchange coupling differs for the effective two-spin model compared to a four-spin model, since it now incorporates the different spin structure of the complexes. Thus, we conclude that the complexes can be effectively treated as spin $\frac{1}{2}$ systems and that they can serve as fundamental building blocks for larger spin structures.’

4. In line 322 the authors mention that the bond could be formed by on-surface chemistry techniques. Have the authors tried this approach? The tip-induced assembly is limited, and this alternative method would increase the impact of the work.

Reply: We thank the reviewer for the suggestion. We have attempted the self-assembly approach by mildly thermal annealing the sample after the co-deposition of Fe atoms and FePc molecules: On the Ag surface, we observed that Fe atoms and molecules become mobile at elevated temperatures, leading to the formation of Fe-FePc complexes without the necessity of tip manipulation [ACS Nano 2025, 19, 1, 1190–1197]. This also worked for another class of molecules (Cu[dbm]₂). On MgO surface, we expect the same process to occur at different annealing conditions. However, we still require isolated Fe atoms for tip preparation, so that this was not pursued in these experiments.

We have now incorporated the citation and description into the manuscript:

'A crucial ingredient is the bond formation between the Fe and the benzene ring, which instead of using tip-assisted assembly could also be created by a thermally driven on-surface self-assembly. This was already successfully demonstrated on metal substrates ([ACS Nano 2025, 19, 1, 1190–1197,⁵⁰) by employing thermal sample annealing. We also believe that these systems could be directly synthesized in fully chemically derived molecular ferrimagnets'.

Reviewer #1 (Remarks to the Author):

The authors addressed all my comments in detail. Only one last issue should be addressed before publication. In their reply, the authors refer to a 'non-polarized calculation' to extract the crystal field. This should be explicitly cited in the figure caption, and the details of this calculation should be added to the computational methods. Is it a restricted open-shell DFT simulation?

Reply: Thank you for your advice. We have now added this in the figure caption and main text: 'In Figure 1d we show the non-spin-polarized projected density of states (PDOS)...'

And we have added the DFT calculation details in the methods. The type of calculation is indeed an Open-shell Restricted Hartree Fock. Open-shell RHF is similar to a non-spin-polarized calculation in DFT where the electronic density is shared, but with an explicit treatment of unpaired electrons. It's a compromise between spin-restricted methods (like RHF) and fully spin-polarized methods (like UHF).

Reviewer #2 (Remarks to the Author):

The authors addressed all my comments. I recommend the manuscript for publication.

Reply: Thank you for your feedback.